# Tunable Electromechanical Coupling Coefficient of a Laterally Excited Bulk Wave Resonator with Composite Piezoelectric Film

**DOI:** 10.3390/mi13040641

**Published:** 2022-04-18

**Authors:** Ying Xie, Yan Liu, Jieyu Liu, Lei Wang, Wenjuan Liu, Bo Woon Soon, Yao Cai, Chengliang Sun

**Affiliations:** 1The Institute of Technological Sciences, Wuhan University, Wuhan 430072, China; _xieying@whu.edu.cn (Y.X.); liuyan92@whu.edu.cn (Y.L.); jieyuuu@whu.edu.cn (J.L.); lei_wang@whu.edu.cn (L.W.); lwjwhu@whu.edu.cn (W.L.); 2Hubei Yangtze Memory Laboratories, Wuhan 430205, China; 3School of Microelectronics, Wuhan University, Wuhan 430072, China; soonbowoon@gmail.com

**Keywords:** composite piezoelectric material, electromechanical coupling coefficient $K_{t}^{2}$, laterally excited bulk wave resonator (XBAR), filter

## Abstract

A resonator with an appropriate electromechanical coupling coefficient ($K_{t}^{2}$) is crucial for filter applications in radio communication. In this paper, we present an effective method to tune the $K_{t}^{2}$ of resonators by introducing different materials into a lithium niobate (LiNbO_3_) piezoelectric matrix. The effective piezoelectric coefficients $e_{33}^{eff}$ and $e_{15}^{eff}$ of composite materials with four different introduced materials were calculated. The results show that the $e_{15}^{eff}$ of SiO_2_/LiNbO_3_ composite piezoelectric material was mostly sensitive to an increase in the width of introduced SiO_2_ material. Simultaneously, the simulation of a laterally excited bulk wave resonator (XBAR) with SiO_2_/LiNbO_3_ composite material was also carried out to verify the change in the $K_{t}^{2}$ originating from the variation in $e_{15}^{eff}$. The achievable n79 filter using the SiO_2_/LiNbO_3_ composite material demonstrates the promising prospects of tuning $K_{t}^{2}$ by introducing different materials into a LiNbO_3_ piezoelectric matrix.

## 1. Introduction

To balance the needs of wide-area coverage and high data rates, 5G new radio (NR) has been proposed [1,2]. Laterally excited bulk acoustic wave resonators (XBAR) are promising candidates for application in fifth-generation mobile communication due to their high frequency, large electromechanical coupling coefficient ($K_{t}^{2}$), low cost and complementary metal oxide semiconductor (CMOS) compatibility [3,4,5,6,7]. Victor Plessky realized a XBAR based on Z-cut lithium niobate (LiNbO_3_) thin plate with a resonance frequency of approximately 4.9 GHz [8]. Ruochen Lu presented first-order antisymmetric (A1) mode resonators in thin 128° Y-cut LiNbO_3_ films with a $K_{t}^{2}$ of 46.4% [9]. Bohua Peng designed and fabricated a solid-mounted-type XBAR on ZY-LiNbO_3_, operating at 5 GHz [5]. The $K_{t}^{2}$ of XBAR has a significant influence on the bandwidth of filters. However, delicate control of the $K_{t}^{2}$ of XBARs is crucial for designing filters; for example, the $K_{t}^{2}$ of LiNbO_3_-based XBARs is too large for specific n79 filters (4.4 GHz–5.0 GHz) [10].

The $K_{t}^{2}$ of XBAR can be adjusted by structural optimization and tuning the piezoelectric coefficients. Gianluca Piazza found that the $K_{t}^{2}$ can be tuned by changing the electrical boundary conditions imposed by the excitation electrodes, obtaining a range varying from 3% to 7% [11]. Jie Zou investigated the influence of the Euler angle and thickness of LiNbO_3_ film on the $K_{t}^{2}$ of the resonator. It was found that the $K_{t}^{2}$ of the A1 mode acoustic wave varied rapidly with changes in the Euler angle [12]. V. Plessky analyzed the influence of pitch and duty factor on frequency and $K_{t}^{2}$ [13]. Using piezoelectric composite materials is another feasible method for $K_{t}^{2}$ tuning. In our previous work, we adopted a ScAlN/AlN composite piezoelectric film to achieve a Lamé Mode resonator with a high $K_{t}^{2}$ of 7.83% [14].

In this paper, we propose an effective method for tuning the $K_{t}^{2}$ of XBARs by applying composite film consisting of LiNbO_3_ piezoelectric material and other materials. The tuning range was as high as 62%, which is efficient compared with other studies, as shown in Table 1. We used FEM to analyze the effective piezoelectric coefficients $e_{33}^{eff}$ and $e_{15}^{eff}$ of composite piezoelectric films with different volume fractions of different materials embedded in LiNbO_3_ piezoelectric material. FEM simulative analysis of XBAR utilizing those composite piezoelectric films was also carried out. Finally, an n79 filter was designed using SiO_2_/LiNbO_3_ composite thin film-based XBARs with an adjustable $K_{t}^{2}$. The proposed XBAR with LiNbO_3_-based composite piezoelectric film shows promising prospects for constructing filters with different bandwidths at high frequency.

## 2. Theoretical Calculation of Piezoelectric Coefficient

The theory of linear piezoelectricity couples the interaction between the electric and elastic variables via the following constitutive equations [15]:(1)$$\sigma_{ij}=C_{ijkl}\varepsilon_{kl}-e_{lij}E_{i}$$
(2)$$D_{i}=e_{ikl}\varepsilon_{kl}+\kappa_{ik}E_{k}$$
where $\varepsilon_{kl}$ and $\sigma_{ij}$ are the components of the elastic strain and the components of the stress tensor, respectively; $E_{i}$ and $D_{i}$ are the components of the electric field and the components of the electric displacement, respectively; $C_{ijkl}$ is the components of the fourth-order elastic stiffness tensor obtained in the absence of an applied electric field; $e_{lij}$ is the components of the piezoelectric modulus tensor obtained without an applied strain; and $\kappa_{ik}$ is the components of the dielectric modulus obtained without an applied strain.

It is convenient to treat the elastic and the electric variables in a similar fashion when modeling the piezoelectric behavior. This is accomplished by employing a notation introduced by Barnett and Lothe [16] and a generalized Voigt two-index notation [17]. Therefore, the constitutive equations can be represented as:(3)$$\left[\begin{array}{c} \sigma \\ D \end{array}\right]=\left[\begin{array}{cc} C & e^{T}\\ e & -\kappa \end{array}\right]\left[\begin{array}{c} \varepsilon \\ -E \end{array}\right]$$

The calculation of the effective properties of composite films is then realized utilizing the homogenization method, which relates the volume-averaged strain, stress, electric field, and electric displacement to the effective properties of the composite film. The composite films can thus be modeled as homogenized media. Using FEM, volume averages can be calculated as follows [18]:(4)$$\bar{\sigma_{ij}}=\frac{1}{V}ʃ\sigma_{ij}dV=\frac{1}{V}\sum_{n=1}^{nel}\sigma_{ij}^{\left(n\right)}V^{\left(n\right)}$$
(5)$$\bar{\varepsilon_{ij}}=\frac{1}{V}ʃ\varepsilon_{ij}dV=\frac{1}{V}\sum_{n=1}^{nel}\varepsilon_{ij}^{\left(n\right)}V^{\left(n\right)}$$
(6)$$\bar{D_{i}}=\frac{1}{V}ʃD_{i}dV=\frac{1}{V}\sum_{n=1}^{nel}D_{i}^{\left(n\right)}V^{\left(n\right)}$$
(7)$$\bar{E_{i}}=\frac{1}{V}ʃE_{i}dV=\frac{1}{V}\sum_{n=1}^{nel}E_{i}^{\left(n\right)}V^{\left(n\right)}$$
where *V* is the volume of the representative volume elements (RVE). $\bar{\sigma_{ij}}$, $\bar{\varepsilon_{ij}}$, $\bar{D_{i}}$, and $\bar{E_{i}}$ are the volume-averaged values of stress, strain, electric displacement, and electric field, respectively.

In terms of these average values, the constitutive equations of linear piezoelectricity for composite material can be expressed in matrix form as follows:(8)$$\begin{array}{ccc} \left\{\begin{array}{c} \bar{\sigma_{11}}\\ \bar{\sigma_{22}}\\ \bar{\sigma_{33}}\\ \bar{\sigma_{23}}\\ \bar{\sigma_{13}}\\ \bar{\sigma_{12}}\\ \bar{D_{1}}\\ \bar{D_{2}}\\ \bar{D_{3}} \end{array}\right\} & =[\begin{array}{ccccccccc} C_{11}^{eff} & C_{12}^{eff} & C_{13}^{eff} & C_{14}^{eff} & 0 & 0 & 0 & -e_{22}^{eff} & e_{31}^{eff}\\ C_{12}^{eff} & C_{11}^{eff} & C_{13}^{eff} & -C_{14}^{eff} & 0 & 0 & 0 & e_{22}^{eff} & e_{31}^{eff}\\ C_{13}^{eff} & C_{13}^{eff} & C_{33}^{eff} & 0 & 0 & 0 & 0 & 0 & e_{33}^{eff}\\ C_{14}^{eff} & -C_{14}^{eff} & 0 & C_{44}^{eff} & 0 & 0 & 0 & e_{15}^{eff} & 0\\ 0 & 0 & 0 & 0 & C_{44}^{eff} & C_{14}^{eff} & e_{15}^{eff} & 0 & 0\\ 0 & 0 & 0 & 0 & C_{14}^{eff} & C_{66}^{eff} & e_{22}^{eff} & 0 & 0\\ 0 & 0 & 0 & 0 & e_{15}^{eff} & e_{22}^{eff} & -\kappa_{11}^{eff} & 0 & 0\\ -e_{22}^{eff} & e_{22}^{eff} & 0 & e_{15}^{eff} & 0 & 0 & 0 & -\kappa_{11}^{eff} & 0\\ e_{31}^{eff} & e_{31}^{eff} & e_{33}^{eff} & 0 & 0 & 0 & 0 & 0 & -\kappa_{33}^{eff} \end{array}] & \left\{\begin{array}{c} \bar{\varepsilon_{11}}\\ \bar{\varepsilon_{22}}\\ \bar{\varepsilon_{33}}\\ 2\bar{\varepsilon_{23}}\\ 2\bar{\varepsilon_{13}}\\ 2\bar{\varepsilon_{12}}\\ -\bar{E_{1}}\\ -\bar{E_{2}}\\ -\bar{E_{3}} \end{array}\right\} \end{array}$$

As shown in Figure 1, the RVE consisted of Z-cut LiNbO_3_ and other materials. Other materials were embedded in the thin LiNbO_3_ film, and the width of the other materials is expressed as *P*. Here, four different materials commonly used in MEMS were taken under consideration, including SiC, Al_2_O_3_, AlN and SiO_2_. The boundary conditions applied to the six surfaces of the RVE are in the form of prescribed displacements and prescribed electric potentials. For calculation of the piezoelectric coefficients $e_{33}^{eff}$ and $e_{15}^{eff}$, the boundary conditions applied to the six surfaces and the postprocessing steps for assessing the piezoelectric coefficients $e_{33}^{eff}$ and $e_{15}^{eff}$ are listed in Table 2. In Table 2, *u*, *v*, and *w* are the displacement components along the *x*-, *y*-, and *z*-coordinate axes, respectively, and *V*0 is the applied electric potential.

The calculated effective piezoelectric coefficients $e_{33}^{eff}$ and $e_{15}^{eff}$ of LiNbO_3_ composites using all four kinds of materials are presented as a function of the width of material (*P*) in Figure 2. The *P* of the other material ranged from zero to a maximum of 19 µm. It is shown that the effective piezoelectric coefficients $e_{33}^{eff}$ and $e_{15}^{eff}$ declined predictably with an increase in *P* for all four kinds of LiNbO_3_-based composite film. Among the four different composite materials, the effective piezoelectric coefficients $e_{33}^{eff}$ and $e_{15}^{eff}$ of the SiO_2_/LiNbO_3_ composite material had the largest variation. The $e_{33}^{eff}$ of AlN/LiNbO_3_ composite film changed the most gradually, while the $e_{15}^{eff}$ of SiC/LiNbO_3_ composite film had the smallest variation. The effective piezoelectric coefficient $e_{15}^{eff}$ of SiO_2_/LiNbO_3_ composite material varied from 3.65 to 1.31 as *P* increased from 0 to 19 µm, for which the tuning range could reach as high as 64.1%.

## 3. FEM Simulation of XBAR

FEM simulation of an XBAR with LiNbO_3_ composite material was also carried out to demonstrate tuning of the $K_{t}^{2}$. As illustrated in Figure 3a, the XBAR consisted of a suspended 300 nm-thick LiNbO_3_ composite platelet and a set of 100 nm-thick Mo electrodes on top. The electrical potentials were alternatingly applied to adjacent electrodes, as illustrated by the “+” and “−” signs in Figure 3b, creating a lateral electric field along the *X* axis. Due to the strong piezoelectric coefficient $e_{15}^{eff}$ of LiNbO_3_, the alternating lateral electric field could excite vertical shear vibration in A1 mode within the platelet [19]. Structural optimization was implemented by adjusting the *P* of the SiO_2_ embedded in the thin LiNbO_3_ film within a range from 0 to 15 µm, while the thickness of SiO_2_ (*t*) remained 150 nm, as shown in Figure 3c.

The series frequency of XBAR with thin LiNbO_3_ film (*p* = 0) is approximately 6.14 GHz and the parallel frequency is 7.25 GHz. As the value of *p* increased, the parallel frequency of XBAR declined consistently, while the series frequency remained almost the same, as shown in Figure 4. The parallel frequency declined from 7.25 GHz to 6.52 GHz as *P* increased from 0 to 15 µm. The series frequency of XBAR can be expressed as the following formula [20]:(9)$$f_{s}=\sqrt{{\left(\frac{v_{z}}{2h}\right)}^{2}+{\left(\frac{v_{x}}{2G}\right)}^{2}}$$
where *h* is the thickness of the piezoelectric thin film and *G* is the gap between two adjacent electrodes. In our simulations, the thickness of the piezoelectric thin film and the gap between two adjacent electrodes remained the same; therefore, it is reasonable to assume that the series frequency remained almost constant. The effective electromechanical coupling coefficient ($K_{t}^{2}$) can be calculated using the following formula [21,22]:(10)$$K_{t}^{2}=\frac{K^{2}}{1+K^{2}}=\frac{\pi^{2}}{4}\times \frac{f_{s}}{f_{p}}\times \frac{\left(f_{p}-f_{s}\right)}{f_{p}}$$
(11)$$K^{2}=\frac{e_{15}^{2}}{\varepsilon_{r}\varepsilon_{0}C_{44}}$$

As shown in Figure 5, when *P* increased to 1 µm, $K_{t}^{2}$ decreased sharply from 32% to 20.7%, and $K_{t}^{2}$ then declined slowly with the increase in *P* from 2 to 11 µm. When *P* increased beyond 11 µm, $K_{t}^{2}$ no longer changed. The variation trend of $K_{t}^{2}$ is highly consistent with the change in the calculated effective piezoelectric coefficient $e_{15}^{eff}$, which demonstrates that introducing other materials to a LiNbO_3_ piezoelectric matrix is an effective method for tuning the $K_{t}^{2}$ of XBARs.

Here, we provide a possible fabrication process flow for our devices, as shown in Figure 6. The substrate wafer consists of a thin Z-cut LiNbO_3_ film and a Si substrate. Firstly, the thin LiNbO_3_ film is etched via electron beam lithography; the depth is controlled by the etching time. A 150 nm-thick layer of SiO_2_ is deposited on the surface of the LiNbO_3_ film and then polished to a smooth plate. Then, molybdenum (Mo) electrodes are deposited on the surface of the thin SiO_2_/LiNbO_3_ film and patterned by lithography and reactive ion etching technology. Subsequently, the release holes are realized via electron beam lithography, which enables formation of the cavity by removing the Si substrate with Xef_2_. By exactly controlling the release time, resonators with only a suspended working area are realized.

## 4. Design of N79 Filters

As discussed in Section 3, the $K_{t}^{2}$ of XBAR can be adjusted by introducing other materials into the LiNbO_3_ piezoelectric film, which enables the construction of different bandwidth filters for 5G. For example, the $K_{t}^{2}$ of an XBAR based on pure LiNbO_3_ film is calculated as being approximately 35% and the −3 dB bandwidth of the corresponding filter is 1050 MHz, as shown in Figure 7a,c, which exceeds the bandwidth requirements of the n79 filter. As seen from Figure 5, the $K_{t}^{2}$ of XBARs decreased to approximately 21% when the *P* of the SiO_2_ in the SiO_2_/LiNbO_3_ composite film was 1 µm, which is suitable for the bandwidth requirement of the n79 filter. Therefore, we designed a filter based on thin SiO_2_/LiNbO_3_ composite film with a *P* of 1 µm. The resonant and anti-resonant frequencies of the series resonator were 4.71 GHz and 5.17 GHz, respectively, and those of the parallel resonator were 4.35 GHz and 4.7 GHz, respectively, as shown in Figure 7b. As shown in Figure 7d, the transmission response of the filter showed a −3 dB bandwidth of 600 MHz, ranging from 4.4 GHz to 5.0 GHz, which satisfies the requirements of the n79 very well.

## 5. Conclusions

In summary, an effective method for tuning the $K_{t}^{2}$ of XBARs, by using composite piezoelectric materials combining LiNbO_3_ piezoelectric material with other materials, is demonstrated in this work. The effective piezoelectric coefficients $e_{33}^{eff}$ and $e_{15}^{eff}$ of the four kinds of LiNbO_3_-based composite materials were calculated through FEM simulation. Among the four different composite materials, the effective piezoelectric coefficients $e_{33}^{eff}$ and $e_{15}^{eff}$ of the SiO_2_/LiNbO_3_ composite material had the largest variation. The $e_{15}^{eff}$ of SiO_2_/LiNbO_3_ composite material declined from 3.65 to 1.31 as *P* increased from 0 to 19 µm. The $e_{33}^{eff}$ of SiO_2_/LiNbO_3_ composite material declined from 1.72 to 0.19 as *P* increased from 0 to 19 µm. Simultaneously, we also carried out the simulation of an XBAR using SiO_2_/LiNbO_3_ composite material to verify the change in the $K_{t}^{2}$, owing to the variation in $e_{15}^{eff}$. The $K_{t}^{2}$ decreased from 34% to approximately 11% as *P* increased from 0 to 17 µm, which was highly consistent with the change in $e_{15}^{eff}$. Finally, we designed a filter made with SiO_2_/LiNbO_3_ composite material, which satisfied the bandwidth requirement of the n79 very well, demonstrating that XBARs with LiNbO_3_-based composite piezoelectric film show fascinating prospects for fabricating different bandwidth filters at high frequencies in the future.

## Figures and Tables

**Figure 1 micromachines-13-00641-f001:**
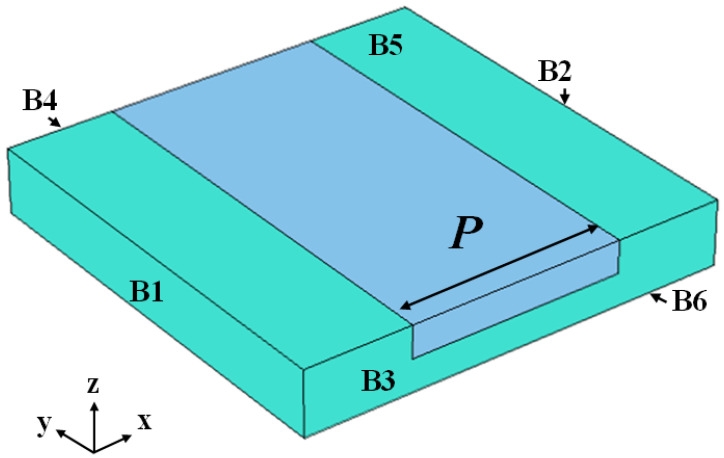
Images of full representative volume elements (RVEs). The green regions represent the LiNbO3 material, whereas the blue region represents the other material.

**Figure 2 micromachines-13-00641-f002:**
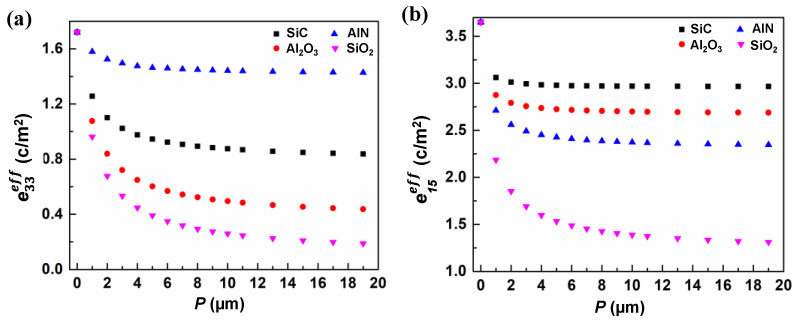
Effective piezoelectric coefficients $e_{33}^{eff}$ (**a**) and $e_{15}^{eff}$ (**b**) of four different LiNbO_3_-based composite materials as function of the width of nonpiezoelectric materials (*P*).

**Figure 3 micromachines-13-00641-f003:**
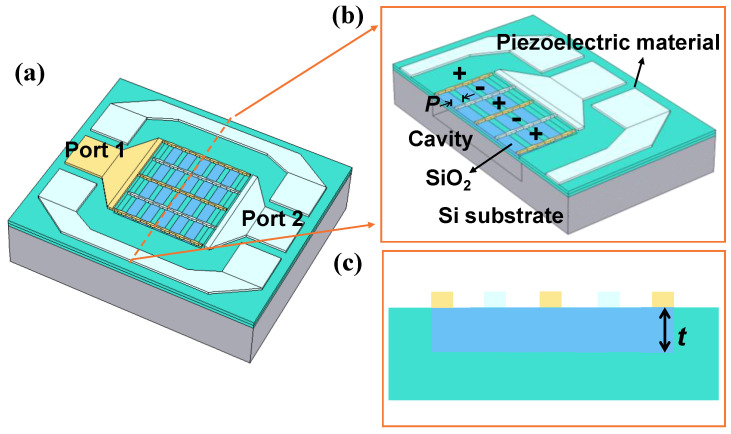
(**a**) Schematic drawing of laterally excited bulk acoustic wave resonator using composite piezoelectric material. (**b**) Sectional view of the resonator cut across the dashed line. (**c**) 2D schematic of the effective working area along the dashed line.

**Figure 4 micromachines-13-00641-f004:**
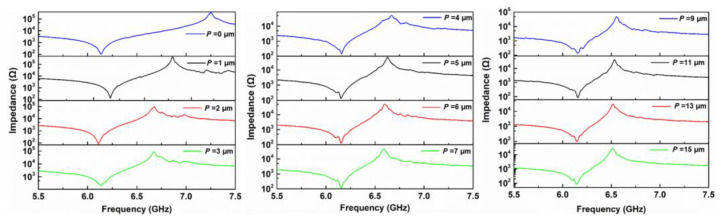
The impedance curves of XBARs with different widths of nonpiezoelectric materials (*P*) ranging from 0 to 15 µm.

**Figure 5 micromachines-13-00641-f005:**
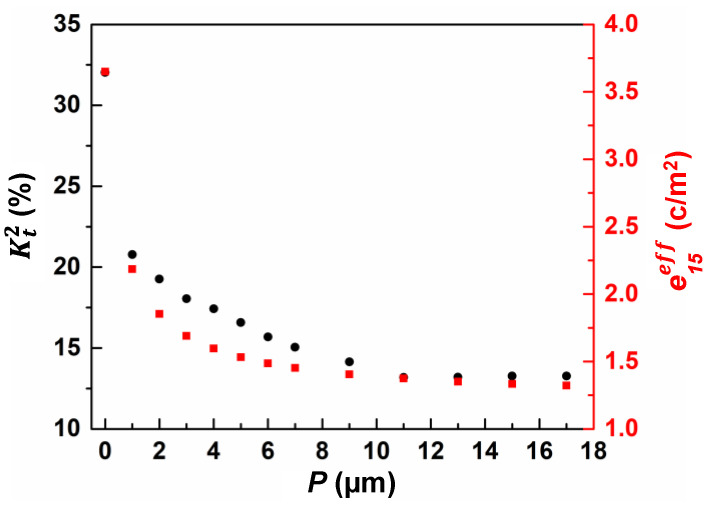
The variation in $K_{t}^{2}$ and $e_{15}^{eff}$ with different widths (*P*) of nonpiezoelectric materials.

**Figure 6 micromachines-13-00641-f006:**
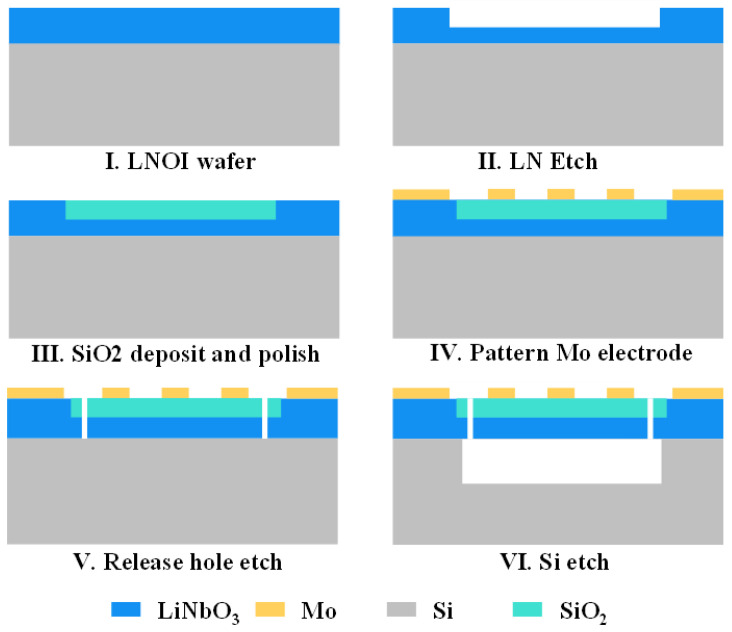
The fabrication process flow for the proposed devices.

**Figure 7 micromachines-13-00641-f007:**
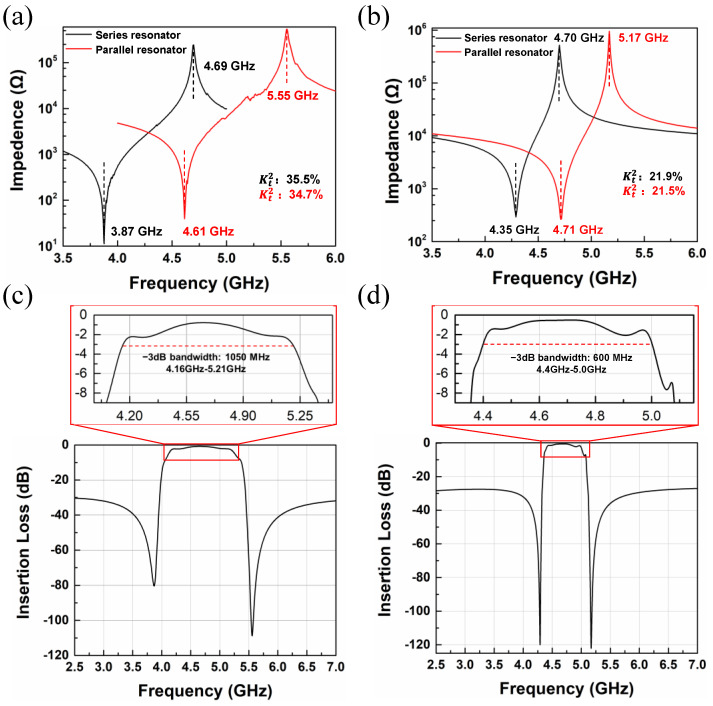
(**a**) The impedance curves of an XBAR with pure LiNbO_3_ film for the n79 filter. (**b**) The impedance curves of an XBAR with SiO_2_/LiNbO_3_ composite film for the n79 filter. (**c**) The response of the proposed filter with pure LiNbO_3_ film. (**d**) The response of the proposed filter with SiO_2_/LiNbO_3_ composite film.

**Table 1 micromachines-13-00641-t001:** Comparison of the tuning effectiveness between our work and previous works.

Ref.	Method	$K_{t}^{2}\;(\%)$	Frequency	Tuning Effect
[12]	Change electrical boundary conditions	19%	484 MHz	10% to 19%
[13]	Change the Euler angle of LiNbO_3_	55%	3.3 GHz	0% to 55%
[14]	Tuning structural parameters	25%	5 GHz	23% to 25%
This work	Composite piezoelectric material	32%	6 GHz	12% to 32%

**Table 2 micromachines-13-00641-t002:** Boundary conditions for evaluating the effective properties of the composite.

Effective Property	B1	B2	B3	B4	B5	B6	Formula
$e_{15}^{eff}$	*u* = 0	*u* = 0	*v* = 0 *φ* = 0	*v* = 0 *φ* = *V*0	*w* = 0	*w* = 0	$e_{15}^{eff}=-\frac{{\bar{\sigma }}_{23}}{{\bar{E}}_{2}}$
$e_{33}^{eff}$	*u* = 0	*u* = 0	*v* = 0	*v* = 0	*w* = 0 *φ* = 0	*w* = 0 φ = *V*0	$e_{33}^{eff}=-\frac{{\bar{\sigma }}_{33}}{{\bar{E}}_{3}}$

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
