# Peer review of "Tunable Electromechanical Coupling Coefficient of a Laterally Excited Bulk Wave Resonator with Composite Piezoelectric Film"

_micromachines, 2022, doi:10.3390/mi13040641_

Round 1

Reviewer 1 Report

The paper demonstrates a mechanism of tuning the coupling coefficient of lithium niobite based XBAR by introducing different materials into lithium niobate (LiNbO3) piezoelectric matrix. The authors demonstrate variation of the effective piezoelectric coefficients with the variation of the width of material introduced into the lithium niobate film mathematically and with FEM simulations. N79 filters are also demonstrated.

In my opinion, the paper is worthy of publication, but it would be good if the authors give clarification and/or corrections based on the following comments and/or suggestions.

Comments/suggestions:

  • The sectional view of the figure presented in Fig. 3(a) marked by the breaking section lines does not coincide with the one in Fig. 3(b).
  • In Fig. 2(a) and (b), the two effective piezoelectric coefficients tend to be constant as the value of P≥10(μm) for silicon dioxide lithium niobite composite. It will be good if the authors give clarification on this.
  • Similar to the comment above, In Fig.5, the coupling coefficient and the effective piezoelectric coefficient show a constant trend with the width (P) of nonpiezoelectric materials from 13(μm) to 19(μm). It will be good if the authors give clarification on this.

Author Response

We thank all the reviewers for carefully reading the manuscript and the constructive comments. For laterally excited bulk wave resonator (XBAR), appropriate electromechanical coupling coefficient () is crucial adjective for filters application in high frequency radio communication. In this work, we present an effective method to tune  of resonators by introducing different materials into lithium niobate (LiNbO3) piezoelectric matrix. In the following pages, we address the reviewers’ comments point by point. In this document, the original reviewers’ comments are in black font; our replies are in blue font. 

Reviewer 2 Report

Authors of the manuscript number “micromachines-1671651” with the title of “Tunable Electromechanical Coupling Coefficient of Laterally Excited Bulk Wave Resonator with Composite Piezoelectric Film” presented a new structural design for XBAR in order to have appropriate electromechanical coupling coefficient. The main application of their device was for 5G and a n79 filter was simulated based on the designed device. The manuscript entails only design and FEM simulation and there was not any measured data or fabricated device in the manuscript. The manuscript has merits for publication in the Micromachines journal, however, there are some major points that authors need to address before publication.

  • The English grammar of the manuscript needs major revision.
  • Paper needs major revision regarding punctuation and citing references.
  • Authors need to provide at least a proposed fabrication process flow for their device in order to give an idea of the fabrication feasibility of their proposed device.
  • They should provide a comparison table in the introduction section to bring out the effectiveness of their design with respect to the previous works.
  • In order to clarify the validity of the issue they are trying to address in this manuscript, they should provide a recent (2022-2021) literature review in the introduction section of the manuscript.
  • In figure 6, what is the reason for having an insertion loss in the range of ⁓ -1 dB for the filter? As Figure 6 is the result of a FEM simulation, why there is a big insertion loss in the response of the filter? Can you please compare this number with previously published papers (simulated and measured results)?

Round 2

Reviewer 2 Report

The manuscript can be accepted for publication in the Micromachines journal.

This manuscript is a resubmission of an earlier submission. The following is a list of the peer review reports and author responses from that submission.